# Nursing students admitted through the affirmative action system display similar performance in professional and academic trajectories to those from the regular path in a public school in Brazil

Marize Lima de Sousa Holanda Biazotto[1,2]*, Leila Bernarda Donato Göttems[1], Fernanda Viana Bittencourt[3], Gilson Roberto de Araújo[2‡], Sérgio Eduardo Soares Fernandes[1‡], Carlos Manoel Lopes Rodrigues[4‡], Francisco de Assis Rocha Neves[2], Fábio Ferreira Amorim[1,2]

1 Graduation Program in Health Sciences, Higher School of Health Sciences (ESCS), Brasília, Federal District, Brazil, 2 Graduation Program in Health Sciences, University of Brasília (UnB), Brasília, Federal District, Brazil, 3 Education and Research Foundation of Health Sciences (FEPECS), Brasília, Federal District, Brazil, 4 Centro Universitário de Brasília (UniCEUB), Brasília, Federal District, Brazil

☯ These authors contributed equally to this work.
‡ GRA, SESF and CMLR also contributed equally to this work.
* marizebiazo@hotmail.com

## Abstract

### Objectives

Affirmative action providing higher education access for socially vulnerable students has been implemented in several countries. However, these policies remain controversial. This study compares the performance of students admitted through the regular path and social quota systems, during and after completion of nursing education, in a public nursing school in Brazil.

### Methods

This retrospective cohort study included all students admitted to nursing school at the School of Health Sciences (ESCS), Brazil, between 2009 and 2014, who were followed until May 2020. The first phase involved document analysis from the ESCS academic management system and Brazilian government agencies. In the second phase, a survey was conducted among the alumni. The social quota system criterion was public school attendance across all primary and secondary education levels.

### Results

Of the 448 students included in the study, 178 (39.7%) were from the affirmative action and 270 (60.3%) from the regular path systems. Affirmative action students were older at the time of nursing school admission (p < 0.001) and took longer to be admitted to the nursing school (p < 0.001) after completing high school. There were no significant differences in the

**Data Availability Statement:** All files used to write the manuscript are available in a public database repository: https://doi.org/10.6084/m9.figshare.14057993.

**Funding:** The authors received no specific funding for this work.

**Competing interests:** The authors have declared that no competing interests exist.

dropout rates and years to complete nursing school. In the second phase, 108 alumni answered the survey. No significant differences were found in their participation in the undergraduate scientific research program and university extension projects, attending residency programs, getting a master's degree and doctoral degree, monthly income, teaching activity, joining public service through a government job competition process, participation in management activities in the private and public health sector, and degree of job satisfaction.

## Conclusion

Our results revealed that affirmative action is a policy that contributes to the reduction of inequalities and guarantees the training of nursing professionals with a similar professional qualification received through affirmative action and regular path systems.

## Introduction

Equality is one of the most established and general principles in the constitutions of most countries worldwide. However, this does not reflect reality since socioeconomic inequality remains a challenge in the global human rights agenda [1, 2]. Furthermore, the disparities between classes are more pronounced in some countries than others [2–5]. In this aspect, Brazil is renowned for its persistent socioeconomic and racial inequality stemming from its history of colonization and slavery [2–8]. The growing recognition of socioeconomic and racial disparities in Brazil has led to public policies to promote social mobility in recent decades, referred to as affirmative action [8–10].

The lack of access to higher education for students from the most vulnerable social classes may perpetuate socioeconomic inequities since it leads them to secondary jobs and hinders social mobility [3, 5, 6, 11–13]. Education level and career choice are essential factors to a student's future socioeconomic condition. In this regard, parents' socioeconomic status primarily impacts their children's educational level and career. High-income families may generate significant investment in their children's education, while students from less privileged socioeconomic classes may experience restrictions that can limit their opportunities [5, 11].

Notably, in Brazil, public universities are tuition free and more prestigious than private ones. Indeed, Brazilian public universities are regarded more highly in all national and international rankings than private universities [4, 14–18]. Thus, undergraduate courses from public universities have higher demand than those from private universities. In this case, the regular path system based only on meritocratic entrance exam scores, traditionally performed in public universities, leads to nearly exclusive admission of higher-income students, since lower-income students show a substantial competitive disadvantage in these exams [6, 8]. This aspect can be explained by the fact that, in contrast to the higher education level, the most exclusive private schools demonstrate better quality at the primary and secondary education levels when compared to public schools [6]. Thus, one way to mitigate this situation and reduce the socioeconomic and racial gaps is affirmative action policies to provide higher education access to undergraduate courses at public universities for students experiencing social vulnerability, further promoting social mobility and social development [3, 6, 8, 12, 13, 19].

Based on the class-based ideas of social inclusion, Brazilian public universities and federal and local governments implemented affirmative action policies to reduce inequality in higher

education access, especially after the 2000s, mainly through social and racial quotas [3, 6, 8, 12, 13, 19]. Social quotas reserve vacancies for candidates with social vulnerability profiles due to low family income, while racial quotas reserve vacancies for racial groups. Social and racial quotas have the same objective: eliminating inequalities and advantages between students from high-income families and those from the most vulnerable social classes [7, 17, 19].

In nursing education, there are particular issues concerning affirmative action. Health systems are rapidly developing and changing, and nurses play an essential role in formulating and implementing health policies, as they comprise half of the global health workforce [20–22]. Nurses' training follows different approaches in each country, arising from their own social, economic, and health contexts, associated with their health and professional training policies [23–25]. In countries with significant social disparities, such as Brazil, it is crucial to train general nurses to consider the reality of the most vulnerable populations and their social and health singularities. These new nursing professionals should face challenges and rapid global transformations. They must understand the complexity of technological advances, competencies, attitudes, and ethics that respond to social demands, health services, and intersectoral actions [26–29]. In this context, nursing students from affirmative action may better understand and embrace underrepresented population groups' health care issues [30–32].

Despite its possible benefits, affirmative action in higher education has recently faced several questions, particularly regarding whether it is only a mechanism of social mobility and whether it really achieves its objectives. Moreover, there are increased issues regarding the possible academic damage incurred due to the admission of less competitive students [6, 10, 13, 14, 33–39]. These arguments are commonly known as the minority mismatch hypothesis. Those who defend this hypothesis argue that affirmative action violates the merit principle, and the students admitted from these policies do not deserve a place in public higher education and cannot succeed in a competitive environment since they are not qualified for undergraduate education. In particular, students admitted from affirmative action are more likely to drop out, take longer to graduate, and perform less well in their future professional performance than regular path students [6, 10, 13, 14, 33–39].

Nurses' availability is crucial for health care systems, including hospitals, long-term care facilities, primary health care services, and home care [21, 22, 40]. The Organization for Economic Co-operation and Development (OECD) points out the need for 9.0 nurses/1,000 inhabitants to ensure access and qualified care to populations. In OECD countries, the average number of nurses/1,000 inhabitants increased from 7.3 in 2000 to 9 in 2015 [28]. The need for nursing staff has led many countries to formulate public policies to intervene in current and future shortages, combined with efforts to promote the retention of these professionals [21, 22, 40]. In some developing countries, such as Colombia, Indonesia, South Africa, India, and Brazil, there are less than 1.5 nurses/1,000 inhabitants [28, 32, 41]. Data from the Brazilian Federal Nursing Council (COFEN) revealed that 582,319 nurses were working in Brazil in March 2021, representing 2.74 nurses/1,000 inhabitants. Despite the increase in the number of nursing courses in Brazil in recent years (from 123 nursing schools in 1997 to 984 in 2017), there is a high attrition among students before graduation, which is among the factors associated with the shortage of nurses. The high number of dropouts from nursing school is even more problematic in tuition-free public universities, where it represents an expense for the government that will not return to the labor force. In this context, the possible higher likelihood of dropping out of affirmative action students compared to other students according to the minority mismatch hypothesis may have even more significant consequences in nursing education [6, 10, 13, 14, 33–39]. Nevertheless, studies performed in Brazil and India showed lower attrition among students admitted via affirmative action than other students across different undergraduate courses [42–44].

Studies evaluating affirmative action students' academic and professional trajectories after finishing their undergraduate courses remain scarce in nursing education. Thus, in an era in which diversity and equality in access to higher education are highly demanded, and given the minority mismatch hypothesis questions regarding affirmative action and student performance, this study aimed to compare the academic and professional trajectories of students admitted through the regular path system with those admitted via the social quota system based on attendance across all primary and secondary education levels in the Federal District's public education system, a surrogate of socioeconomic status.

## Materials and methods

This retrospective cohort study included all students admitted to the undergraduate School of Health Sciences (ESCS), Brasília, Federal District, Brazil, between 2009 and 2014, involving a retrospective follow-up of the cohort through May 2020. The ESCS is a public undergraduate educational institution established in 2001 under the State Department of Health of the Federal District, Brazil. It is structured based on the Brazilian Unified Health System (SUS) principles, offering undergraduate medical and nursing courses free of charge [45, 46]. The curriculum is common to all students and based on active teaching and learning methodologies in the Federal District's public health system (a system-based curriculum) [45–47].

The ESCS nursing school was opened in 2009. The students' entrance selection is conducted yearly through the Brazilian Unified Selection System for Undergraduate Education Institutions (SISU) based on students' scores on the Brazilian National High School Exam (ENEM). The ENEM is a national-level exam performed to assess students' readiness for admission in undergraduate courses, similar to the Scholastic Aptitude Test (SAT) in the United States [48]. The ESCS nursing school has used the social quota-based affirmative action admission system from the beginning. The criterion is attendance across all primary and secondary education levels in the Federal District's public education system, a proxy for socioeconomic status, and its aim is to surpass the limited participation of minorities in the nursing profession [49]. Each year, 80 students are admitted to the ESCS nursing school based only on their best grades on the ENEM. Thirty-two vacancies are reserved for students who have completed all primary and secondary education levels in the Federal District's public education system (social quota system), and 48 vacancies are offered by the traditional admission system (regular path system). Notably, the minimum score for admission in the regular path system is higher than for the social quota system. Thus, all students who meet the criteria for the social quota system apply to it. This aspect is also observed in other public universities that adopt affirmative policies [6, 8].

The study inclusion criterion was being a student admitted to nursing school at the ESCS between 2009 and 2014. The exclusion criteria were (1) students who were still attending nursing school by the end of the follow-up (May 2020), and (2) students admitted by mandatory transfers from other higher education institutions determined by law (relatives of Brazilian public and military servants). Of the 449 students admitted between 2009 and 2014, one was excluded for not completing the course at the end of follow-up (0.2%); thus, the study included 448 students.

In the first phase (June 2019–May 2020), data collection was conducted through document analysis from the following sources in Brazil: ESCS academic management system; the Brazilian researchers' curriculum directory from the Lattes Platform, managed by the National Council for Scientific and Technological Development (CNPq - http://buscatextual.cnpq.br/buscatextual/busca.do); the Graduation Programs Database on the Sucupira Platform, maintained by the Coordination for the Improvement of Higher Education Personnel

(CAPES - https://sucupira.capes.gov.br/sucupira/public/index.jsf), Ministry of Education; the Federal Nursing Council (COFEN), and the Official Gazette publications of the Government of Federal District, Brazil.

In this phase, the variables collected included demographic characteristics, Human Development Index (HDI) and its dimensions (decent standard of living, long and healthy life, and access to education) in the place of residence at the time of nursing school admission, per capita income and average household income of the place of residence at time of nursing school admission, year of completion of high school, year of the commencement and completion of nursing school, dropping out before finishing the course, and the admission path (regular path or the social quota system). These variables were used primarily to test the minority mismatch hypothesis that students from affirmative action may be more likely to drop out and take longer to graduate (the minimum period to finish nursing school is four years) [6, 10, 13, 14, 33–39].

In the second phase (May 2020–June 2020), we distributed a survey to nursing alumni to collect additional data regarding their trajectories during nursing school and academic and professional activities after undergraduate education. These variables were used to test the minority mismatch hypothesis that affirmative action students may perform worse in their academic and future professional performance than regular path students [6, 10, 13, 14, 33–39]. The survey was sent to all ESCS nursing school alumni admitted between 2009 and 2014 (n = 253); 108 responses were received, accounting for 42.7% of the total nursing graduates sent the survey. Of the 108 alumni who responded in the second phase, 99 provided complete responses, while nine did not answer about remuneration per hour and the degree of job satisfaction. No responses were excluded.

In the survey, the variables collected to access the alumni's trajectories during nursing school included participation in a scientific research program, university extension, or academic mentoring for other students, as well as feeling confident to work upon finishing nursing school. The variables used to access the alumni's academic and professional trajectories after nursing school included job and management activity in the public and private health sector, residency program attendance, teaching activity, monthly income, remuneration per hour, achievement of MSc and Ph.D., and degree of job satisfaction (on a 5-point Likert scale). The survey also collected demographic characteristics, whether the alumni received a scholarship for students experiencing social vulnerability during nursing school, and year of the commencement and completion of nursing school.

The variables' distribution and normality were checked using histograms, dispersion diagrams, and the Kolmogorov–Smirnov and Shapiro-Wilk tests. Quantitative data were expressed as mean and standard deviation (SD) or median and interquartile range (IQR: 25–75%). Categorical variables were expressed as numbers or percentages. The student's t-test or Mann-Whitney test was used to compare quantitative variables, as appropriate. Contingency tables were used for categorical variables, and Pearson's chi-square test ($\chi2$) or Fisher's exact test was used as necessary. To evaluate the variables that determine performance at graduation, binary logistic regression and probit regression models were built, and multivariate analysis was performed. The following binary dependent variables (coded as 0 or 1) were considered: (i) attrition from the nursing school and (iii) nursing school completion over four years.

To analyze the dependent variables in the survey questions, a propensity score matching for social quota system admission was performed applying a logit regression model adjusted for sex and time after nursing school completion was used. The EZR software version 1.54 (Saitama Medical Center, Jichi Medical University, Japan) with a 1:1 pair-matching ratio without replacement on the logit of the propensity score was employed in this analysis using a caliper of 0.2 width.

Statistical analyses were performed using IBM SPSS (version 23.0 for Mac), GRETL (version 2021b for MS Windows), statistical software R version 4.0.5 (R Foundation for Statistical Computing,) and EZR software version 1.54 (Saitama Medical Center, Jichi Medical University, Japan). Statistical significance was set at $p < 0.05$.

The Ethics Committee of the Education and Research Foundation of Health Sciences (FEPECS), Brasília, Federal District, Brazil, approved this study (21324719.1.3001.5553). For the study's first phase, the ethics committee waived the requirement for informed consent due to the retrospective study design. For the second phase, written informed consent was obtained from all alumni.

## Results

The study included 448 students; 60.3% (270/448) were from the regular path system and 39.7% (178/448) from the social quota system. As presented in Table 1, the median age was 18.0 years (IQR: 17.0–21.0 years), 78.8% (353/448) were female, 10.7% were married (48/448), and the median time (in years) between high school completion and admission to nursing school was 1.0 (IQR: 0.0–3.0) years. Of all students admitted during the study period, 43.5% (195/448) dropped out of nursing school. Among the students who completed nursing school,

**Table 1. Univariate analysis comparing students admitted to nursing school at the School of Health Sciences (ESCS), Brasília, Federal District, Brazil, from the regular path and social quota systems between 2009 and 2014 (n = 448).**

| Variable | Total | Regular Path System | Social Quota System | p-value |
|---|---|---|---|---|
| | (n = 448) | (n = 270) | (n = 178) | |
| Age at admission, years, median (IQR 25–75%) | 18.0 (17.0–21.0) | 18.0 (17.0–19.0) | 19.0 (18.0–25.0) | < 0.001 |
| Female gender, n (%) | 353 (78.8) | 220 (81.5) | 133 (74.7) | 0.087 |
| Married, n (%) | 48 (10.7) | 19 (7.0) | 29 (16.3) | 0.002 |
| HDI of the place of residence at admission, median (IQR 25–75%)[a] | 0.830 (0.748–0.905) | 0.866 (0.809–0.953) | 0.808 (0.726–0.853) | < 0.001 |
| Decent standard of living | 0.830 (0.725–0.918) | 0.870 (0.785–1.000) | 0.780 (0.698–0.841) | < 0.001 |
| Long and healthy life | 0.905 (0.851–0.922) | 0.916 (0.887–0.934) | 0.886 (0.827–0.908) | < 0.001 |
| Access to education | 0.792 (0.693–0881) | 0.814 (0.741–0.914) | 0.740 (0.666–0.795) | < 0.001 |
| Per capita income of the place of residence at admission, MW, median (IQR 25–75%)[a] | 1.775 (0.933–2.725) | 1.998 (1.396–3.391) | 0.983 (0.915–1.998) | < 0.001 |
| Average household income of the place of residence at admission, MW, median (IQR 25–75%)[a] | 5.596 (3.183–7.997) | 6.072 (4.445–9.494) | 3.239 (3.101–6.072) | < 0.001 |
| Time between high school completion and admission to nursing school, years, median (IQR 25–75%) | 1.0 (0.0–3.0) | 1.0 (0.0–2.0) | 2.0 (0.0–7.2) | < 0.001 |
| Attrition, n (%) | 195 (43.5) | 120 (44.4) | 75 (42.1) | 0.629 |
| Time taken to complete nursing school, years, median (IQR 25–75%)[b] | 4.0 (4.0–4.0) | 4.0 (4.0–4.0) | 4.0 (4.0–4.0) | 0.837 |
| Nursing school completion above four years, n (%)[b] | 35 (13.8) | 21 (14.0) | 14 (13.6) | 0.926 |

HDI, Human Development Index; MW, Brazilian minimum wage in Brazilian Real; SD, standard deviation; IQR, interquartile range.

[a]n = 438: students who were admitted to the nursing school between 2009 and 2014 and stated their place of residence at time of nursing school admission; of these, 264 were from the regular path system and 174 from the social quota system; 10 students did not state their place of residence at the time of nursing school admission.

[b]n = 253: students who were admitted to the nursing school between 2009 and 2014 and graduated; of these, 150 were from the regular path system and 103 from the social quota system; 195 students dropped out of nursing school.

**Table 2. Multivariate analysis of nursing school attrition and nursing school completion in more than four years among students admitted to nursing school at the School of Health Sciences (ESCS), Brasília, Federal District, Brazil, between 2009 and 2014.**

| Variables | Collinearity Statistics | | Binary Logistic Regression | | | Probit Regression | |
|---|---|---|---|---|---|---|---|
| | Tolerance | VIF | Slope | OR (95% CI) | p-value | Slope | p-value |
| **Attrition (events = 195)[a]** | | | | | | | |
| Marital status (married versus single) | 0.826 | 1.210 | -0.0750 | 0.729 (0.361–1.472) | 0.378 | -0.0750 | 0.362 |
| Gender (female versus male) | 0.984 | 1.016 | -0.1656 | 0.511 (0.319–0.818) | 0.005 | -0.1656 | 0.005 |
| Age at the admission (per one year) | 0.174 | 5.758 | 0.0159 | 1.068 (0.981–1.163) | 0.132 | 0.0160 | 0.132 |
| Admission system (social quota system versus regular path system) | 0.821 | 1.218 | -0.0613 | 0.776 (0.500–1.205) | 0.259 | -0.0614 | 0.261 |
| Time between high school completion and admission to nursing school, years (per one year) | 0.171 | 5.849 | -0.0053 | 0.978 (0.883–1.084) | 0.675 | -0.0053 | 0.674 |
| Average household income of the place of residence at admission, median (per one Brazilian minimum wage) | 0.882 | 1.134 | -0.0008 | 0.997 (0.950–1.046) | 0.895 | -0.0008 | 0.897 |
| **Nursing school completion above four years (events = 195)[b]** | | | | | | | |
| Marital status (married versus single) | 0.802 | 1.247 | 0.1035 | 2.112 (0.675–6.620) | 0.199 | 0.1016 | 0.215 |
| Gender (female versus male) | 0.965 | 1.036 | 0.0694 | 2.112 (0.639–6.973) | 0.220 | 0.6878 | 0.217 |
| Age at the admission (per one year) | 0.133 | 7.500 | 0.0234 | 1.234 (1.045–1.456) | 0.013 | 0.0260 | 0.011 |
| Admission system (social quota system versus regular path system) | 0.789 | 1.267 | -0.0232 | 0.810 (0.350–1.875) | 0.623 | -0.0250 | 0.604 |
| Time between high school completion and admission to nursing school, years (per one year) | 0.128 | 7.831 | -0.0257 | 0.795 (0.646–0.977) | 0.029 | -0.0282 | 0.025 |
| Average household income of the place of residence at admission, median (per one Brazilian minimum wage) | 0.868 | 1.152 | -0.0015 | 0.986 (0.905–1.075) | 0.756 | -0.0011 | 0.822 |

VEF, variance inflation factor; SE, standard error; OR, odds ratio; CI, confidence interval.

[a]Binary logistic model–Hosmer-Lemeshow test: $\chi^2$ = 6.592; df = 8; p-value = 0.581; Probit model–Normality test: $\chi^2$ = 1.156; p-value = 0.468

[b]Binary logistic model–Hosmer-Lemeshow test: $\chi^2$ = 10.617; df = 8; p-value = 0.224; Probit model–Normality test: $\chi^2$ = 5.525; p-value = 0.063

the median time to completion was 4.0 years (IQR: 4.0–4.0 years), with 13.8% (35/254) of the students completing in over four years.

Table 1 compares the sociodemographic characteristics and academic trajectories between the students admitted through the regular path and social quota systems. Students from the social quota system were older at the time of admission to the nursing school (p < 0.001), and more of them were married (p = 0.002). The place of residence of social quota students at the time of nursing school admission had lower HDI (p < 0.001), decent standard of living–HDI dimension (p < 0.001), long and healthy life–HDI dimension (p < 0.001), access to education–HDI dimension (p < 0.001), per capita income (p < 0.001), and average household income (p < 0.001) than that of regular path students. Furthermore, they spent more time between high school and access to nursing school (p < 0.001). There was no significant difference regarding nursing school attrition (p = 0.629), the time taken to complete nursing school (p = 0.837), and students taking more than four years to complete the nursing school (p = 0.926).

Table 2 presents the multivariate analysis of nursing school attrition and students taking more than four years to complete nursing school. The type of admission system was not

**Table 3. Comparison between all alumni (n = 448) and the students who responded to the survey (n = 108).**

| Variable | Total | | | Regular Path System | | | Social Quota System | | |
|---|---|---|---|---|---|---|---|---|---|
| | All students (n = 448) | Survey students (n = 108) | p-value | All students (n = 270) | Survey students (n = 70) | p-value | All students (n = 178) | Survey students (n = 38) | p-value |
| Age at admission, years, median (IQR 25–75%) | 18.0 (17.0–21.0) | 18.0 (17.0–19.0) | 0.077 | 18.0 (17.0–19.0) | 18.0 (17.0–19.0) | 0.429 | 18.0 (17.0–19.0) | 18.0 (17.8–20.2) | 0.104 |
| Female gender, n (%) | 353 (78.8) | 82 (75.9) | 0.517 | 220 (81.5) | 56 (80.0) | 0.778 | 133 (74.7) | 26 (68.4) | 0.424 |
| Married, n (%) | 48 (10.7) | 6 (5.6) | 0.104 | 19 (7.0) | 1 (1.4) | 0.076 | 29 (16.3) | 5 (13.2) | 0.630 |
| Time taken to complete nursing school, years, median (IQR 25–75%) | 4.0 (4.0–4.0)[a] | 4.0 (4.0–4.0) | 0.486 | 4.0 (4.0–4.0)[a] | 4.0 (4.0–4.0) | 0.970 | 4.0 (4.0–4.0)[a] | 4.0 (4.0–4.2) | 0.232 |

IQR, interquartile range; SD, standard deviation.

[a]n = 253: students who were admitted to the nursing school between 2009 and 2014 and graduated; of these, 150 were from the regular path system and 103 from the social quota system; 195 students dropped out of nursing school.

independently associated with nursing school attrition (OR, 0.776; 95% CI, 0.500–1.205; p = 0.259) and nursing school completion in more than four years (OR, 0.810; 95% CI, 0.350–1.875; p = 0.623). The only variable independently associated with nursing school attrition was gender, as women were less likely to drop out before completing nursing school (OR, 0.511; 95% CI, 0.319–0.818; p = 0.005). Age (OR, 1.234; 95% CI, 1.045–1.456; p = 0.013) and time between high school completion and admission to nursing school (OR, 0.795; 95% CI, 0.646–0.977; p = 0.029) were independently associated with nursing school completion in more than four years.

In total, 108 students completed the survey. There were no significant differences between all alumni and the students who responded to the survey regarding age at nursing school admission, gender, marital status, and time taken to complete nursing school (Table 3).

Table 4 compares the survey responses between students admitted from the regular path and social quota systems. Students from the social quota system received more scholarships for social vulnerability than regular path students (p < 0.001). There were no significant differences in academic activities during nursing school, such as participation in undergraduate scientific research programs (p = 0.629), academic mentoring to other students during undergraduate courses (p = 0.253), and attending university extension programs (p = 0.670). In addition, after completion of nursing school, there was no significant difference between the alumni from the two systems, including attending residency (p = 0.183), MSc (p = 0.722), and Ph.D. (p = 0.579) programs, monthly income (p = 0.565), teaching activity (p = 0.719), joining public service through a government job competition process (p = 0.897), participation in management activities in the private and public health sector (p = 0.829 and p = 0.705, respectively), and degree of job satisfaction (p = 0.297). These results did not change after propensity score matching for social quota system admission adjusted for sex and time after nursing school completion.

## Discussion

Brazil implemented affirmative action policies mainly after the 2000s to reduce inequalities in higher education access [3, 6, 8, 12, 13, 19]. While nursing education has been extensively researched, few studies have evaluated affirmative action in nursing schools. Furthermore, most of the studies to evaluate affirmative action in higher education have focused on academic outcomes during the undergraduate course. Few studies have analyzed students' academic and professional performance after graduation [19]. In our study, which compared

**Table 4. Univariate analysis and analysis with propensity score matching for social quota system admission adjusted for sex and time after nursing school completion comparing students admitted from the regular path and social quota systems in survey questions (n = 108).**

| Variable | Total (n = 108) | Social Quota System (n = 38) | Regular Path System (n = 70) | p-value | Regular Path System (n = 38)[a] | p-value[a] |
|---|---|---|---|---|---|---|
| Living in the Federal District, Brazil, after nursing school completion, n (%) | 103 (95.4) | 36 (94.7) | 67 (95.7) | 0.817 | 32 (97.0) | 0.555 |
| Receive scholarship for students experiencing social vulnerability, n (%) | 28 (25.9) | 27 (71.1) | 1 (1.4) | < 0.001 | 1 (3.0) | < 0.001 |
| Participation in undergraduate scientific research program, n (%) | 45 (41.7) | 15 (39.5) | 30 (42.9) | 0.733 | 12 (36.4) | 1.000 |
| Academic mentoring to other students, n (%) | 33 (30.6) | 9 (31.6) | 24 (27.1) | 0.253 | 15 (45.2) | 0.071 |
| Attending to university extension project, n (%) | 31 (28.7) | 12 (31.6) | 19 (27.1) | 0.626 | 7 (21.2) | 0.398 |
| Attending to a residency program, n (%) | 66 (61.1) | 20 (52.6) | 46 (65.7) | 0.183 | 21 (63.6) | 0.614 |
| Master's degree, n (%) | 12 (11.1) | 4 (10.5) | 8 (11.4) | 0.877 | 3 (9.1) | 0.689 |
| Doctor's degree, n (%) | 5 (4.6) | 2 (5.3) | 3 (4.3) | 0.579 | 1 (3.0) | 0.555 |
| Feeling confident to work at the finish of the nursing school, n (%) | 101 (93.5) | 34 (89.5) | 67 (95.7) | 0.208 | 32 (97.0) | 0.163 |
| Teaching activity, n (%) | 18 (16.7) | 7 (18.4) | 11 (15.7) | 0.719 | 3 (9.1) | 0.282 |
| Joined public service through a government job competition process, n (%) | 22 (20.4) | 8 (21.1) | 14 (20.0) | 0.897 | 7 (21.2) | 1.000 |
| Management activity in the public health sector, n (%) | 7 (6.5) | 2 (5.3) | 5 (7.1) | 0.705 | 1 (3.0) | 0.555 |
| Job in the private health sector, n (%) | 41 (38.0) | 13 (34.2) | 28 (40.0) | 0.554 | 6 (18.2) | 0.757 |
| Management activity in the private health sector, n (%) | 24 (22.2) | 8 (21.1) | 16 (22.9) | 0.829 | 11 (33.3) | 0.609 |
| Monthly income, MW, median (IQR 25–75%)[b] | 7.2 (2.9–7.2) | 7.2 (2.9–7.2) | 7.2 (2.9–7.2) | 0.603 | 2.9 (2.9–7.2) | 0.310 |
| Remuneration per hour, R$/hour, median (IQR 25–75%)[b] | 136.4 (66.7–214.3) | 136.4 (60.6–214.3) | 151.5 (66.7–214.3) | 0.540 | 166.7 (79.2–214.3) | 0.305 |
| Degree of job satisfaction, 0 to 5, median (IQR 25–75%)[b] | 4.0 (3.0–5.0) | 4.0 (3.0–5.0) | 4.0 (3.0–5.0) | 0.297 | 4.0 (3.0–5.0) | 0.066 |

MW, Brazilian minimum wage in Brazilian Real; R$, Brazilian Real; IQR, interquartile range; SD, standard deviation.

[a]Analysis with propensity score matching for social quota system admission adjusted for sex and time taken after nursing school.

[b]Nine alumni did not answer.

students admitted to nursing school via the social quota system and those admitted through the regular path system, the social quota system contributed to decreasing socioeconomic inequality by admitting students with reduced socioeconomic status. Furthermore, there were no significant differences in academic and professional trajectories between the social quota and regular path students during and after nursing school, which contradicts the minority mismatch hypothesis [6, 10, 13, 14, 33–39].

Regarding socioeconomic background, the social quota students lived in places with lower per capita income, average household income, and HDI than regular path students. Furthermore, in the survey sample, 27 of 37 social quota students in nursing school received a scholarship for students experiencing social vulnerability that aims to retain and assist students with monthly household income per capita up to two Brazilian minimum wages. These findings indicate that the social quota system selected the most vulnerable socioeconomic students who likely experience barriers to attending public universities with only highly competitive traditional admission systems [6]. Indeed, the capability of affirmative action to promote equality in higher education by redistributing university admission slots for the most vulnerable students has been illustrated in different studies, whether by racial or social quota [6, 8, 50]. At the University of Brasília, Brazil, Black students from the racial quota system reported lower family incomes and parenteral educational levels and higher attendance at a public secondary

school than students from the regular admission system [6, 8]. In India, a study analyzing the quota system for underrepresented castes in an engineering school observed that this affirmative action significantly improves admission of the poorest students [50].

The role of the social quota system as an instrument to surpass the socioeconomic status boundaries in access to nursing education was also demonstrated by the higher age of the affirmative action students compared to regular path students at attendance, as well as the increased duration between completing high school and attending nursing school. In the Federal District, Brazil, before the ESCS's nursing school inauguration in 2009, there was only one undergraduate nursing course offered by a public university. Indeed, the increase in Brazilian nursing schools in recent decades was mainly due to the positions provided by private universities limiting access to students with better economic conditions [27, 51]. The few vacancies available in the public education system make it difficult for students from less privileged social classes to access higher nursing schools. This difficulty may explain why, in the present study, the quota students were older than non-quota students. This finding may, in fact, be favorable since previous studies have reported a positive effect of higher age on nursing school admission. In particular, an Australian study explored age as an indicator of academic achievement in nursing education and found that students over 25 years were more successful than those under 25 years [52]. Similarly, a UK study observed that students over 25 years performed better than those aged 21–25 years [53].

The main objective of affirmative action in higher education is to promote social mobility [3, 6, 8, 12, 13, 19]. Regarding this objective, one study found that successful applicants for the racial quota admission system at the University of Brasília had more years of schooling seven to eight years after the exam than quota applicants who failed the entrance exam, with a score ranked above the 85th percentile in the entire applicant pool. The successful quota application also increased their likelihood of working as a director or manager [19]. Furthermore, there is a shortage in the nursing workforce in Brazil and other countries, especially in the more socially vulnerable regions [21, 22, 40]. Affirmative action can help address this challenge. Despite these findings, there has been massive debate regarding the consequences of the cultural and educational knowledge mismatch between quota and non-quota students before university attendance in the academic environment and the students' educational and job market performance [6, 10, 13, 14, 33–39].

Regarding educational performance during nursing school, we did not observe a significant difference in the time necessary to complete the undergraduate program between quota and regular path students. This finding challenges the hypothesis of the educational commitment in quota students since, in this situation, reduced performance would be expected in these students' evaluation tests and, consequently, an increased time to graduate. A study at the University of Brasília also demonstrated similar performance between quota and non-quota students in nursing school [54]. In a larger Brazilian study that also evaluated students from public nursing schools, the authors did not observe a significant difference between the affirmative action and regular path students in performance on the Brazilian National Exam taken at the end of their courses [17]. Indeed, other Brazilian studies have observed that racial quotas improved the educational outcomes for the intended disadvantaged group, as quota students performed as well as or better than non-quota students [8, 55, 56].

Our study did not find a significant difference in attrition between affirmative action and regular path students. Furthermore, there was no significant difference between the quota and non-quota students in academic activities during nursing school, such as undergraduate scientific research programs, academic mentoring to other students, and university extension programs. These results contradict the hypothesis that affirmative action students may have higher attrition and lower performance in educational activities than non-quota students due

to their unfavorable cultural and educational gaps [6, 10, 13, 14, 33–39]. Previous studies in Brazil across different undergraduate courses have found even better results for quota students, presenting lower attrition than regular path students [42, 44, 56]. Nevertheless, other studies evaluating affirmative action in the United States, Israel, and India have refuted that affirmative action adversely impacts dropout rates in higher education [43, 57–60].

Studies evaluating the professional trajectory of affirmative action students after completing their undergraduate courses remain scarce. In our study, there were no significant differences in academic and professional trajectories between these students after completing nursing school—including attending residency programs, achievement of MSc and Ph.D., and job placement or participation in management activities in the Brazilian public and private health sectors—with both groups showing similar monthly incomes, job satisfaction, employability, and participation in management activities in the Brazilian public and private health sectors. A study conducted among alumni admitted to undergraduate programs through affirmative action initiatives at Malaysian public universities reported difficulties finding employment in occupations that matched their qualifications, even within affirmative action in public sector employment [61]. The differences in the status of public universities in Malaysia and Brazil may explain these conflicting results. In Brazil, public universities have more prestigious and better scientific production [4, 16–18], facilitating their students' employment after graduation, unlike in Malaysia, where private universities are better rated than public ones [61].

Previous studies have also examined the performance of affirmative action students and their subsequent gains after graduation, generally finding that the earnings of affirmative action students tend to outweigh the potential costs raised by the minority mismatch hypothesis [8, 62–64]. Moreover, a study conducted in India found that alumni from an engineering school admitted through a social quota system for underrepresented castes presented substantial benefits in the labor market after eight to ten years, despite their lower entrance exam scores [50]. A Brazilian study found that, based on an administrative dataset collected by the Brazilian Ministry of Labor that covers about 97% of all formal workers in Brazil, social quota students from the University of Brasília, mainly males, had increased years of education, college completion, and labor earnings [19]. Furthermore, a study in the United States found that returns on attending a selective college are large for Hispanics, Blacks, and those whose parents are relatively poorly educated [65]. That is, students with the greatest social vulnerability were the ones who had the highest earnings after attending undergraduate programs. Indeed, increased access to higher education via affirmative action may be an effective mechanism to promote intergenerational mobility, leading the students to achieve levels of socioeconomic status above those of their parents [5, 11].

Our study has several limitations. First, we evaluated students from only one nursing school in Brazil; thus, further extensive studies that include other universities are needed to confirm our results. Moreover, as documentary data were used in the first phase, some relevant variables for evaluation among affirmative action and regular path students could not be obtained. In addition, the second phase included only students who completed the online survey, which may selectively include an increased proportion of students interested in the study topic. Notably, our study results should be interpreted in the context of nursing education within a Brazilian public higher education institution with its specificities and characteristics, such as the high standing of public universities in Brazil. Other variables, like the regional level of unemployment, were not evaluated in the survey but may influence one's ability to obtain a new job in the public and private sector and the monthly income in the alumni's place of residence. However, most of the ESCS's nursing school alumni remained in the Federal District after completion of nursing school (95.4%); therefore, in theory, they were exposed to a similar set of geographical variables that may interfere in the studied outcomes. Our study also did not

evaluate the opportunity cost of economic inequality and the effects of the COVID-19 pandemic on nursing alumni's professional and academic performance, which should be assessed in future studies. Finally, we evaluated the academic and professional trajectories of students who have completed nursing school up to a maximum of eight years. Thus, future studies should examine the long-term impact of the social quota system on alumni's academic and professional outcomes.

## Conclusion

Our findings showed that affirmative action is a policy that contributes to reducing inequalities and guarantees the training of nursing professionals with similar professional and academic trajectories during and after nursing school. Factors such as the time taken to complete nursing school, attrition, employment, monthly income, and job satisfaction were similar between social quota and regular path students. These results contradict the minority mismatch hypothesis and reinforce the role of affirmative action as an instrument to mitigate inequalities and promote social mobility.

## Acknowledgments

We would like to thank Editage (www.editage.com) for English language editing.

## Author Contributions

**Conceptualization:** Marize Lima de Sousa Holanda Biazotto, Leila Bernarda Donato Göttems, Fernanda Viana Bittencourt, Gilson Roberto de Araújo, Sérgio Eduardo Soares Fernandes, Carlos Manoel Lopes Rodrigues, Francisco de Assis Rocha Neves, Fábio Ferreira Amorim.

**Data curation:** Marize Lima de Sousa Holanda Biazotto, Leila Bernarda Donato Göttems, Fernanda Viana Bittencourt, Francisco de Assis Rocha Neves, Fábio Ferreira Amorim.

**Formal analysis:** Marize Lima de Sousa Holanda Biazotto, Leila Bernarda Donato Göttems, Fernanda Viana Bittencourt, Gilson Roberto de Araújo, Sérgio Eduardo Soares Fernandes, Carlos Manoel Lopes Rodrigues, Francisco de Assis Rocha Neves, Fábio Ferreira Amorim.

**Investigation:** Marize Lima de Sousa Holanda Biazotto, Leila Bernarda Donato Göttems, Fernanda Viana Bittencourt, Francisco de Assis Rocha Neves, Fábio Ferreira Amorim.

**Methodology:** Marize Lima de Sousa Holanda Biazotto, Leila Bernarda Donato Göttems, Fernanda Viana Bittencourt, Francisco de Assis Rocha Neves, Fábio Ferreira Amorim.

**Project administration:** Marize Lima de Sousa Holanda Biazotto, Leila Bernarda Donato Göttems, Fernanda Viana Bittencourt, Francisco de Assis Rocha Neves, Fábio Ferreira Amorim.

**Resources:** Marize Lima de Sousa Holanda Biazotto, Leila Bernarda Donato Göttems, Fernanda Viana Bittencourt, Gilson Roberto de Araújo, Sérgio Eduardo Soares Fernandes, Carlos Manoel Lopes Rodrigues, Francisco de Assis Rocha Neves, Fábio Ferreira Amorim.

**Software:** Marize Lima de Sousa Holanda Biazotto, Leila Bernarda Donato Göttems, Fernanda Viana Bittencourt, Francisco de Assis Rocha Neves, Fábio Ferreira Amorim.

**Supervision:** Marize Lima de Sousa Holanda Biazotto, Leila Bernarda Donato Göttems, Fernanda Viana Bittencourt, Francisco de Assis Rocha Neves, Fábio Ferreira Amorim.

**Validation:** Marize Lima de Sousa Holanda Biazotto, Leila Bernarda Donato Göttems, Fernanda Viana Bittencourt, Francisco de Assis Rocha Neves, Fábio Ferreira Amorim.

**Visualization:** Marize Lima de Sousa Holanda Biazotto, Leila Bernarda Donato Göttems, Fernanda Viana Bittencourt, Gilson Roberto de Araújo, Sérgio Eduardo Soares Fernandes, Carlos Manoel Lopes Rodrigues, Francisco de Assis Rocha Neves, Fábio Ferreira Amorim.

**Writing – original draft:** Marize Lima de Sousa Holanda Biazotto, Leila Bernarda Donato Göttems, Fernanda Viana Bittencourt, Francisco de Assis Rocha Neves, Fábio Ferreira Amorim.

**Writing – review & editing:** Gilson Roberto de Araújo, Sérgio Eduardo Soares Fernandes, Carlos Manoel Lopes Rodrigues.

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
