## [Decision Letter · Decision Letter 0]

13 Jul 2021

PONE-D-21-11217

Nursing students admitted through the affirmative action system display a similar performance in professional and academic trajectories as those from the regular path in a public school in Brazil

PLOS ONE

Dear Prof. De Sousa Holanda Biazotto,

Thank you for submitting your manuscript to PLOS ONE. After careful consideration, we feel that it has merit but does not fully meet PLOS ONE’s publication criteria as it currently stands. Therefore, we invite you to submit a revised version of the manuscript that addresses the points raised during the review process.

Each reviewer has engaged with the paper in a substantial and considerate way. Each of them raises some concerns about the analysis and its presentation. The reviewers’ comments are numerous and require thoughtful and detailed work. I am confident that if you engage with all the reviewers’ suggestions in a thoughtful and cogent way, the quality of your paper will be definitely enhanced.

We look forward to receiving your revised manuscript.

Kind regards,

Giuseppe Lucio Gaeta, Ph.D

Academic Editor

PLOS ONE

Additional Editor Comments (if provided):

Reviewers' comments:

Reviewer's Responses to Questions

**Comments to the Author**

1. Is the manuscript technically sound, and do the data support the conclusions?

Reviewer #1: Partly

Reviewer #2: Yes

2. Has the statistical analysis been performed appropriately and rigorously? 

Reviewer #1: No

Reviewer #2: Yes

4. Is the manuscript presented in an intelligible fashion and written in standard English?

Reviewer #1: No

Reviewer #2: Yes

5. Review Comments to the Author

Reviewer #1: This article is focused on a relevant and emerging field – The role of Education Policies at higher levels for reducing inequalities, especially at degree’s achievements.

There are promising sections in the paper but there are also opportunities of improvement. Let me address the most claiming of these weak sections:

Point 1: The English style must be revised for the entire text. The Review of Literature must be redone, considering a different path. For instances, I suggest Authors to divide the Review of Literature considering the different dimensions interfering in the evolution of each different dimension of socio-economic inequality and latter to focus on the importance of reducing inequalities at school/university degrees’achievements.

Authors must then elaborate and justify the rationale behind the set of variables surveyed in the models.

Additionally, authors must discuss better their arguments. There is not a clear linkage the Review of Literature and the empirical effort (e.g., which literature did suggest the studied dimensions in questionnaire?).

Additionally, authors must consider the possibility of several other methods for robustness (mostly anticipated in their review of literature - analysis of discriminatory models, canonical correlation methods, structural equation models, probit and ordered probit dimensions, etc)

Point 2: Tables and Figures can be improved in the presented format. There are also different types/styles of edition which must be fixed.

There are additional insights which must be discussed again. Authors must try to detail the rationale behind some possible ‘exogenous’ variables, like the geographical dummies, taking the opportunity to include additional dimensions like the regional level of unemployment (coeteris paribus, a proxy for the difficulties of getting a new job if hired/proxy for latent demand) and a dimension related to the opportunity cost of economic inequality). It became unclear whether Authors cannot use a panel data/longitudinal analysis by taking an additional attention on these data. Considering the time of data collection, additional insights, namely the effects of pandemic situation, must be discussed on particular biases on responses.

Point 3. Finally, I will also appreciate to have more information about the implications, weaknesses and challenges of the data sources. Authors shall also try standard methods for binary data (namely probit x logit comparison as well as exhibit marginal effects for the variables on Table 3). Considering Table 3, several problems of multicollinearity and of endogeneity must be discussed, as well as the robustness of the estimated errors considering heteroscedasticity. In Table 4, the p-value related to the t- or z- tests must be detailed, given the different samples’ sizes.

Point 4. Discussion of Results as well as Conclusions must be revised (in length and in content). Besides an English revision, the text posits sentences which have not been even tested along the previous text: “In Brazil, there is persistently high racial and socioeconomic inequality due to its history of colonization and slavery.” Or “The lack of access to higher education for students 260from the most vulnerable social classes may perpetuate socioeconomic inequities by 261hindering social mobility”. For this 2nd quote, I suggest some reads namely articles like Gomes et al (2021, International Journal of Manpower) or Hout (2018, PNAS).

Reviewer #2: “Nursing students admitted through the affirmative action system display a similar performance in professional and academic trajectories as those from the regular path in a public school in Brazil”

Research question: To compare the outcomes (academic, employment, and professional) of nursing students admitted through regular and quota systems at a public undergraduate institution in Brazil.

Findings: Examining comprehensive data on students admitted during 2009-2014, the paper finds that students admitted through the quota system were somewhat older. However, there were no significant differences in the likelihood of completing nursing school. Examining survey data on students admitted during 2009-2014, the paper finds there were no significant differences in earnings, public sector employment, job satisfaction, as well as other outcomes.

Comments/suggestions:

1. Thanks for writing this paper. It is important to understand more about affirmative action in public health careers.

2. It would improve the paper to define more precisely what is meant by affirmative action in education. Public policies vary a lot across the world and even in Brazil. Sometimes they are based on race/ethnicity, sometimes on socioeconomic status, and sometimes on both. The paper should clarify in the abstract and introduction that the policy under consideration is one related to public school attendance, a proxy for socioeconomic status.

3. The paper assumes (rightly) there are differences in socioeconomic status between regular and quota nursing students but doesn’t actually demonstrate this. If the data exist, it would improve the paper to compare the socioeconomic background of students, e.g., private/public school attendance, parents' education. It would especially be interesting to know what percentage of regular students had attended private primary and secondary school.

4. It appears there are errors in Table 2. The row on nursing school completion above four years says that 14 regular students and 21 quota students took longer than four years to complete school. However, if 150 regular students and 103 quota students graduated, then the percentage taking more than four years should be 9.3% for regular students and 20.4% for quota students. Please correct the errors in the table and check the related multivariate results in Table 3.

5. Given the survey response rate was about 43%, it would be useful to investigate how the survey participants were different from the student population. This analysis can be done separately for normal and quota students and can be an “appendix” table. Such an analysis would help to contextualize the interpretation of results in Table 4.

6. Reference #2 (Francis and Tannuri-Pianto) is relevant. They have other work that is even more relevant. One paper (Francis, Andrew M. and Maria Tannuri-Pianto. 2012. "Using Brazil’s Racial Continuum to Examine the Short-Term Effects of Affirmative Action in Higher Education." Journal of Human Resources, 47(3): 754-784.) examines differences in academic performance between regular and quota students at the University of Brasilia. Another paper (Francis-Tan, Andrew and Maria Tannuri-Pianto. 2018. "Black Movement: Using discontinuities in admissions to study the effects of college quality and affirmative action." Journal of Development Economics, 135: 97-116.) examines the effect of quotas on college completion and labor earnings.

---

## [Author Response · Author response to Decision Letter 0]

19 Sep 2021

September 19, 2021

Emily Chenette

Editor-in-Chief

PLOS ONE

Dear Editor:

I would like to resubmit the attached manuscript titled “Nursing students admitted through the affirmative action system display similar performance in professional and academic trajectories to those from the regular path in a public school in Brazil.” The manuscript ID is PONE-D-21-11217.

The manuscript has been carefully rechecked and the appropriate changes have been made in accordance with the referees’ suggestions. The changes have been highlighted in yellow for the convenience of the reviewers. Responses to the reveiwers’ comments have been prepared and are included below.

We thank the referees for their thoughtful suggestions and insights, which have enriched the manuscript and produced a more balanced and better account of the research. We hope that the revised manuscript is now suitable for publication in PLOS ONE.

We look forward to your reply.

Sincerely,

Marize Lima de Sousa Holanda Biazotto

Graduation Program in Health Sciences, University of Brasília (UnB), Brasília, Federal District, Brazils

Faculdade de Medicina Faculdade de Ciências de Saúde Campos Univ. Darcy Ribeiro s/n - Asa Norte, Brasília - DF, 70910-900 

Phone: +55 61999639496

Email: marizebiazo@hotmail.com

Response to reviewers

Reviewer #1: 

This article is focused on a relevant and emerging field – The role of Education Policies at higher levels for reducing inequalities, especially at degree’s achievements.

There are promising sections in the paper but there are also opportunities of improvement. 

Let me address the most claiming of these weak sections:

Comment 1: The English style must be revised for the entire text. 

Response: Thank you for the comment. The English style was revised by an academic translation service. 

Comment 2: The Review of Literature must be redone, considering a different path. For instances, I suggest Authors to divide the Review of Literature considering the different dimensions interfering in the evolution of each different dimension of socio-economic inequality and latter to focus on the importance of reducing inequalities at school/university degrees’ achievements.

Response: We thank you for the comment that showed us that we need to strengthen some aspects of our manuscript. We made changes in the Introduction to clarify our study's scientific background and rationale, mainly the importance of reducing inequalities in school/university degree achievement (including the intergenerational occupational legacy), the affirmative action policies in nursing education, and the questions aimed at affirmative action policies (mainly the minority mismatch hypothesis).

The changes are quoted below:

“Equality is one of the most established and general principles in the constitutions of most countries worldwide. However, this does not reflect reality since socioeconomic inequality remains a challenge in the global human rights agenda [1,2]. Furthermore, the disparities between classes are more pronounced in some countries than others [2-5]. In this aspect, Brazil is renowned for its persistent socioeconomic and racial inequality stemming from its history of colonization and slavery [2-8]. The growing recognition of socioeconomic and racial disparities in Brazil has led to public policies to promote social mobility in recent decades, referred to as affirmative action [8-10].

The lack of access to higher education for students from the most vulnerable social classes may perpetuate socioeconomic inequities since it leads them to secondary jobs and hinders social mobility [3,5,6,11-13]. Education level and career choice are essential factors to a student's future socioeconomic condition. In this regard, parents' socioeconomic status primarily impacts their children's educational level and career. High-income families may generate significant investment in their children's education, while students from less privileged socioeconomic classes may experience restrictions that can limit their opportunities [5,11]. 

Notably, in Brazil, public universities are tuition free and more prestigious than private ones. Indeed, Brazilian public universities are regarded more highly in all national and international rankings than private universities [4,14-18]. Thus, undergraduate courses from public universities have higher demand than those from private universities. In this case, the regular path system based only on meritocratic entrance exam scores, traditionally performed in public universities, leads to nearly exclusive admission of higher-income students, since lower-income students show a substantial competitive disadvantage in these exams [6,8]. This aspect can be explained by the fact that, in contrast to the higher education level, the most exclusive private schools demonstrate better quality at the primary and secondary education levels when compared to public schools [6]. Thus, one way to mitigate this situation and reduce the socioeconomic and racial gaps is affirmative action policies to provide higher education access to undergraduate courses at public universities for students experiencing social vulnerability, further promoting social mobility and social development [3,6,8,12,13,19].”

Comment 3: Authors must then elaborate and justify the rationale behind the set of variables surveyed in the models. Additionally, authors must discuss better their arguments. There is not a clear linkage the Review of Literature and the empirical effort (e.g., which literature did suggest the studied dimensions in questionnaire?).

Response: We thank you for the comment. We have clarified the rationale behind the aspects in the Introduction, especially the linkage between the studied dimensions in our questionnaire and our study aim to analyze the questions against the affirmative action raised by the minority mismatch hypothesis, such as whether affirmative action students had higher dropout rates, took longer to graduate, or performed lower in their future professional lives than regular-path students. 

The changes are quoted below:

“Despite its possible benefits, affirmative action in higher education has recently faced several questions, particularly regarding whether it is only a mechanism of social mobility and whether it really achieves its objectives. Moreover, there are increased issues regarding the possible academic damage incurred due to the admission of less competitive students [6,10,13,14,33-39]. These arguments are commonly known as the minority mismatch hypothesis. Those who defend this hypothesis argue that affirmative action violates the merit principle, and the students admitted from these policies do not deserve a place in public higher education and cannot succeed in a competitive environment since they are not qualified for undergraduate education. In particular, students admitted from affirmative action are more likely to drop out, take longer to graduate, and perform less well in their future professional performance than regular path students [6,10,13,14,33-39].”

“(…) in an era in which diversity and equality in access to higher education are highly demanded, and given the minority mismatch hypothesis questions regarding affirmative action and student performance, this study aimed to compare the academic and professional trajectories of students admitted through the regular path system with those admitted via the social quota system based on attendance across all primary and secondary education levels in the Federal District's public education system, a surrogate of socioeconomic status.”

Comment 4: Additionally, authors must consider the possibility of several other methods for robustness (mostly anticipated in their review of literature - analysis of discriminatory models, canonical correlation methods, structural equation models, probit and ordered probit dimensions, etc)

Response: We thank you for the observation. We redid the logit analysis, including average household income of the place of residence at admission as an independent variable, and included the probit analysis in Table 3 of the original manuscript (Table 2 of the revised manuscript). We also included a propensity score matching for social quota system admission adjusted for sex and time taken after nursing school completion in the analysis of dependent variables in the survey questions in Table 4 of the original manuscript (Table 4 of the revised manuscript).

The changes are quoted below:

- Material and methods – “To evaluate the variables that determine performance at graduation, binary logistic regression and probit regression models were built, and multivariate analysis was performed.”

“A propensity score matching for social quota system admission adjusted for sex and time after nursing school completion was used to analyze the dependent variables in the survey questions.”

Comment 5: Tables and Figures can be improved in the presented format. There are also different types/styles of edition which must be fixed.

Response: We thank you for the observation. We reviewed and updated the tables in the revised manuscript.

Comment 6: There are additional insights which must be discussed again. Authors must try to detail the rationale behind some possible ‘exogenous’ variables, like the geographical dummies, taking the opportunity to include additional dimensions like the regional level of unemployment (coeteris paribus, a proxy for the difficulties of getting a new job if hired/proxy for latent demand) and a dimension related to the opportunity cost of economic inequality. It became unclear whether Authors cannot use a panel data/longitudinal analysis by taking an additional attention on these data. Considering the time of data collection, additional insights, namely the effects of pandemic situation, must be discussed on particular biases on responses.

Response: We thank you for the observation. In this respect, we clarify that most of the ESCS’ nursing school alumni remain in the Federal District after completing their undergraduate program. In this case, possible geographical dummies that may interfere in the studied outcomes, such as the regional level of unemployment, are similar for most alumni. The study of a dimension related to the opportunity cost of economic inequality was not the aim of our study, but it is an excellent suggestion for future studies.

For a better understanding of these aspects, we included the following sentences in the Discussion (limitations of the study):

“Other variables, like the regional level of unemployment, were not evaluated in the survey but may influence one’s ability to obtain a new job in the public and private sector and the monthly income in the alumni’s place of residence. However, most of the ESCS’s nursing school alumni remained in the Federal District after completion of nursing school (95.4%); therefore, in theory, they were exposed to a similar set of geographical variables that may interfere in the studied outcomes. Our study also did not evaluate the opportunity cost of economic inequality and the effects of the COVID-19 pandemic on nursing alumni's professional and academic performance, which should be assessed in future studies. Finally, we evaluated the academic and professional trajectories of students who have completed nursing school up to a maximum of eight years. Thus, future studies should examine the long-term impact of the social quota system on alumni’s academic and professional outcomes.”

Comment 7: Finally, I will also appreciate to have more information about the implications, weaknesses and challenges of the data sources. Authors shall also try standard methods for binary data (namely probit x logit comparison as well as exhibit marginal effects for the variables on Table 3). Considering Table 3, several problems of multicollinearity and of endogeneity must be discussed, as well as the robustness of the estimated errors considering heteroscedasticity. 

Response: We thank you for the observation. The Collinearity Statistics and Hosmer‐Lemeshow Test have been included in Table 3. All independent variables showed tolerance above 0.100 and VIF under 10.00. We also reconducted the logit analysis, including average household income of the place of residence at admission as an independent variable and including the marginal effects. We also performed the probit analysis and have included it in Table 3.

Comment 8: In Table 4, the p-value related to the t- or z- tests must be detailed, given the different samples’ sizes.

Response: We included a propensity score matching for social quota system admission adjusted for sex and time taken after nursing school completion in the analysis of dependent variables in the survey questions in Table 4 of the original manuscript (Table 4 of the revised manuscript). The results did not change.

Comment 9: Discussion of Results as well as Conclusions must be revised (in length and in content). Besides an English revision, the text posits sentences which have not been even tested along the previous text: “In Brazil, there is persistently high racial and socioeconomic inequality due to its history of colonization and slavery.” Or “The lack of access to higher education for students from the most vulnerable social classes may perpetuate socioeconomic inequities by hindering social mobility”. For this 2nd quote, I suggest some reads namely articles like Gomes et al (2021, International Journal of Manpower) or Hout (2018, PNAS).

Response: We thank you for the observation and the suggested articles. We revised the Discussion and Conclusions. Regarding the sentences pointed out by the reviewer, we moved them to the literature review in the Introduction. The English style was revised.

We appreciate the manuscripts of Gomes et al. (2021) and Hout et al. (2018), which examine the intergenerational occupational legacy, and have included them in our references, as quoted below:

- Introduction – “Education level and career choice are essential factors to a student's future socioeconomic condition. In this regard, parents' socioeconomic status primarily impacts their children's educational level and career. High-income families may generate significant investment in their children's education, while students from less privileged socioeconomic classes may experience restrictions that can limit their opportunities [5,11]” 

- Discussion – “A Brazilian study found that, based on an administrative dataset collected by the Brazilian Ministry of Labor that covers about 97% of all formal workers in Brazil, social quota students from the University of Brasília, mainly males, had increased years of education, college completion, and labor earnings [19]. Furthermore, a study in the United States found that returns on attending a selective college are large for Hispanics, Blacks, and those whose parents are relatively poorly educated [65]. That is, students with the greatest social vulnerability were the ones who had the highest earnings after attending undergraduate programs. Indeed, increased access to higher education via affirmative action may be an effective mechanism to promote intergenerational mobility, leading the students to achieve levels of socioeconomic status above those of their parents [5,11].”

Reviewer #2: 

“Nursing students admitted through the affirmative action system display similar performance in professional and academic trajectories to those from the regular path in a public school in Brazil”

Research question: To compare the outcomes (academic, employment, and professional) of nursing students admitted through regular and quota systems at a public undergraduate institution in Brazil.

Findings: Examining comprehensive data on students admitted during 2009-2014, the paper finds that students admitted through the quota system were somewhat older. However, there were no significant differences in the likelihood of completing nursing school. Examining survey data on students admitted during 2009-2014, the paper finds there were no significant differences in earnings, public sector employment, job satisfaction, as well as other outcomes.

Comments/suggestions:

Comment 1: Thanks for writing this paper. It is important to understand more about affirmative action in public health careers.

Response: Many thanks for the nice comment.

Comment 2: It would improve the paper to define more precisely what is meant by affirmative action in education. Public policies vary a lot across the world and even in Brazil. Sometimes they are based on race/ethnicity, sometimes on socioeconomic status, and sometimes on both. The paper should clarify in the abstract and introduction that the policy under consideration is one related to public school attendance, a proxy for socioeconomic status.

Response: We thank you for the comment. We made changes in the Introduction to define more precisely affirmative action in education and its importance in reducing inequalities, including a sentence differentiating the criteria of social and racial quota systems. Furthermore, we have clarified in the Abstract, Introduction, and Methods that the policy under consideration is related to public school attendance.

The changes are quoted below:

- Abstract – “The social quota system criterion was public school attendance across all primary and secondary education levels.”

- Introduction – “Based on the class-based ideas of social inclusion, Brazilian public universities and federal and local governments implemented affirmative action policies to reduce inequality in higher education access, especially after the 2000s, mainly through social and racial quotas [3,6,8,12,13,19]. Social quotas reserve vacancies for candidates with social vulnerability profiles due to low family income, while racial quotas reserve vacancies for racial groups. Social and racial quotas have the same objective: eliminating inequalities and advantages between students from high-income families and those from the most vulnerable social classes [7,17,19].”

- Introduction – “(…) this study aimed to compare the academic and professional trajectories of students admitted through the regular path system with those admitted via the social quota system based on attendance across all primary and secondary education levels in the Federal District's public education system, a surrogate of socioeconomic status.”

- Methods – “The ESCS nursing school has used the social quota-based affirmative action admission system from the beginning. The criterion is attendance across all primary and secondary education levels in the Federal District's public education system, a proxy for socioeconomic status, and its aim is to surpass the limited participation of minorities in the nursing profession [49]”.

Comment 3: The paper assumes (rightly) there are differences in socioeconomic status between regular and quota nursing students but doesn’t actually demonstrate this. If the data exist, it would improve the paper to compare the socioeconomic background of students, e.g., private/public school attendance, parents' education. It would especially be interesting to know what percentage of regular students had attended private primary and secondary school.

Response: We thank you for this concern. There was no student in the regular path system that attended all primary and secondary levels in private primary and secondary schools. As the students can apply to only one of the two systems (social quota or affirmative action system), and the minimum score for admission is higher in the regular path system than social quota system, all students that met the criteria for the social quota system applied to it. In addition, to compare the socioeconomic status between regular and quota nursing students, we also compared the Human Development Index (HDI) and its dimensions (decent standard of living, long and healthy life, and access to education), as well as per capita income and average household income of place of residence between regular path and social quota students at the time of medical school admission. Furthermore, in the survey phase, we included a comparison between regular path and social quota students regarding the scholarship for students experiencing social vulnerability. 

The changes are quoted below:

- Methods – “The ESCS nursing school has used the social quota-based affirmative action admission system from the beginning. The criterion is attendance across all primary and secondary education levels in the Federal District's public education system, a proxy for socioeconomic status, and its aim is to surpass the limited participation of minorities in the nursing profession [49]. Each year, 80 students are admitted to the ESCS nursing school based only on their best grades on the ENEM. Thirty-two vacancies are reserved for students who have completed all primary and secondary education levels in the Federal District's public education system (social quota system), and 48 vacancies are offered by the traditional admission system (regular path system). Notably, the minimum score for admission in the regular path system is higher than for the social quota system. Thus, all students who meet the criteria for the social quota system apply to it. This aspect is also observed in other public universities that adopt affirmative policies [6,8].”

- Methods – “In this phase, the variables collected included demographic characteristics, Human Development Index (HDI) and its dimensions (decent standard of living, long and healthy life, and access to education) in the place of residence at the time of nursing school admission, per capita income and average household income of the place of residence at time of nursing school admission, year of completion of high school, year of the commencement and completion of nursing school, dropping out before finishing the course, and the admission path (regular path or the social quota system).”

“In the survey, the variables collected to access the alumni’s trajectories during nursing school included participation in a scientific research program, university extension, or academic mentoring for other students, as well as feeling confident to work upon finishing nursing school. The variables used to access the alumni’s academic and professional trajectories after nursing school included job and management activity in the public and private health sector, residency program attendance, teaching activity, monthly income, remuneration per hour, achievement of MSc and Ph.D., and degree of job satisfaction (on a 5-point Likert scale). The survey also collected demographic characteristics, whether the alumni received a scholarship for students experiencing social vulnerability during nursing school, and year of the commencement and completion of nursing school.”

- Results – “The place of residence of social quota students at the time of nursing school admission had lower HDI (p < 0.001), decent standard of living – HDI dimension (p < 0.001), long and healthy life – HDI dimension (p < 0.001), access to education – HDI dimension (p < 0.001), per capita income (p < 0.001), and average household income (p < 0.001) than that of regular path students.”

“Students from the social quota system received more scholarships for social vulnerability than regular path students (p < 0.001).”

Comment 4: It appears there are errors in Table 2. The row on nursing school completion above four years says that 14 regular students and 21 quota students took longer than four years to complete school. However, if 150 regular students and 103 quota students graduated, then the percentage taking more than four years should be 9.3% for regular students and 20.4% for quota students. Please correct the errors in the table and check the related multivariate results in Table 3.

Response: We thank you for this concern. We corrected the percentages of regular students to 14.0% (21/150) and quota students to 13.6% (14/103). We also reconducted the logit analysis including average household income of the place of residence at admission as an independent variable.

Comment 5: Given the survey response rate was about 43%, it would be useful to investigate how the survey participants were different from the student population. This analysis can be done separately for normal and quota students and can be an “appendix” table. Such an analysis would help to contextualize the interpretation of results in Table 4.

Response: We thank you for this suggestion. We included the comparison between all alumni and the students who responded to the survey (Table 3 of reviewed version). There were no significant differences between all alumni and the students that responded to the survey regarding age at nursing school admission, gender, marital status, and time taken to complete nursing school. We also included an analysis of propensity score matching for social quota system admission adjusted for sex and time taken after nursing school completion in the analysis of dependent variables in the survey questions in Table 4 of the original manuscript (Table 4 of the revised manuscript).

Comment 6: Reference #2 (Francis and Tannuri-Pianto) is relevant. They have other work that is even more relevant. One paper (Francis, Andrew M. and Maria Tannuri-Pianto. 2012. "Using Brazil’s Racial Continuum to Examine the Short-Term Effects of Affirmative Action in Higher Education." Journal of Human Resources, 47(3): 754-784.) examines differences in academic performance between regular and quota students at the University of Brasilia. Another paper (Francis-Tan, Andrew and Maria Tannuri-Pianto. 2018. "Black Movement: Using discontinuities in admissions to study the effects of college quality and affirmative action." Journal of Development Economics, 135: 97-116.) examines the effect of quotas on college completion and labor earnings.

Response: Many thanks for the suggested papers. We appreciate the manuscripts of Francis et al. (2012) and Francis et al. (2018). We have included these references in our paper:

“Indeed, the capability of affirmative action to promote equality in higher education by redistributing university admission slots for the most vulnerable students has been illustrated in different studies, whether by racial or social quota [6,8,50]. At the University of Brasília, Brazil, Black students from the racial quota system reported lower family incomes and parenteral educational levels and higher attendance at a public secondary school than students from the regular admission system [6,8].”

“(…) a study showed that successful applicants for the racial quota admission system at the University of Brasília had more years of schooling seven to eight years after the exam than quota applicants who failed the entrance exam, with a score ranked above the 85th percentile in the entire applicant pool. The successful quota application also increased their likelihood of working as a director or manager [19].”

---

## [Decision Letter · Decision Letter 1]

24 Jan 2022

PONE-D-21-11217R1Nursing students admitted through the affirmative action system display similar performance in professional and academic trajectories to those from the regular path in a public school in BrazilPLOS ONE

Dear Dr. Biazotto,

Thank you for submitting your manuscript to PLOS ONE. After careful consideration, we feel that it has merit but does not fully meet PLOS ONE’s publication criteria as it currently stands. Therefore, we invite you to submit a revised version of the manuscript that addresses the points raised during the review process.

More specifically, the reviewers recommend reconsideration of your manuscript following minor revision and modification.Please submit your revised manuscript by Mar 10 2022 11:59PM. If you will need more time than this to complete your revisions, please reply to this message or contact the journal office at plosone@plos.org. Please include the following items when submitting your revised manuscript:A rebuttal letter that responds to each point raised by the academic editor and reviewer(s). You should upload this letter as a separate file labeled 'Response to Reviewers'.A marked-up copy of your manuscript that highlights changes made to the original version. You should upload this as a separate file labeled 'Revised Manuscript with Track Changes'.An unmarked version of your revised paper without tracked changes. You should upload this as a separate file labeled 'Manuscript'.If applicable, we recommend that you deposit your laboratory protocols in protocols.io to enhance the reproducibility of your results. Protocols.io assigns your protocol its own identifier (DOI) so that it can be cited independently in the future. For instructions see: https://journals.plos.org/plosone/s/submission-guidelines#loc-laboratory-protocols. Additionally, PLOS ONE offers an option for publishing peer-reviewed Lab Protocol articles, which describe protocols hosted on protocols.io. Read more information on sharing protocols at https://plos.org/protocols?utm_medium=editorial-email&utm_source=authorletters&utm_campaign=protocols.

We look forward to receiving your revised manuscript.

Kind regards,

Giuseppe Lucio Gaeta, Ph.D

Academic Editor

PLOS ONE

Reviewers' comments:

Reviewer's Responses to Questions

**Comments to the Author**

1. If the authors have adequately addressed your comments raised in a previous round of review and you feel that this manuscript is now acceptable for publication, you may indicate that here to bypass the “Comments to the Author” section, enter your conflict of interest statement in the “Confidential to Editor” section, and submit your "Accept" recommendation.

Reviewer #1: (No Response)

Reviewer #2: (No Response)

2. Is the manuscript technically sound, and do the data support the conclusions?

Reviewer #1: Partly

Reviewer #2: Yes

3. Has the statistical analysis been performed appropriately and rigorously? 

Reviewer #1: No

Reviewer #2: Yes

4. Have the authors made all data underlying the findings in their manuscript fully available?

Reviewer #1: Yes

Reviewer #2: Yes

5. Is the manuscript presented in an intelligible fashion and written in standard English?

Reviewer #1: Yes

Reviewer #2: Yes

6. Review Comments to the Author

Reviewer #1: There was an interesting revision of the paper. However, Authors introduced now the relevant element of Propensity Scores matching with the expected details. I urge Authors to fix this minor issue.

Reviewer #2: Thanks for revising the paper extensively. The changes made the paper more insightful and more convincing. The only issue that I'd like to raise is related to Table 1 (page 12).

I think the errors in the table are still uncorrected. The row on nursing school completion above four years says that 14 regular students and 21 quota students took longer than four years to complete school. However, if 150 regular students and 103 quota students graduated, then the percentage taking more than four years should be 9.3% for regular students and 20.4% for quota students.

The errors concern either the percentages OR the numbers in the table note. If the percentages, then also update/correct the statement on page 11 that "there was no significant difference regarding ... the time taken to complete nursing school (p = 0.837)."

7. PLOS authors have the option to publish the peer review history of their article (what does this mean?). If published, this will include your full peer review and any attached files.

Reviewer #1: No

Reviewer #2: No

---

## [Author Response · Author response to Decision Letter 1]

10 Feb 2022

We thank the reviewers for their thoughtful suggestions and insights, which have enriched the manuscript and produced a more balanced and better account of the research.

Reviewer #1: 

There was an interesting revision of the paper. However, Authors introduced now the relevant element of Propensity Scores matching with the expected details. I urge Authors to fix this minor issue.

Response: Thank you for the comment and careful appraisal of the study. We clarify the description of the propensity score-matching in Methods as quoted below.

“To analyze the dependent variables in the survey questions, a propensity score matching for social quota system admission was performed applying a logit regression model adjusted for sex and time after nursing school completion was used. The EZR software version 1.54 (Saitama Medical Center, Jichi Medical University, Japan) with a 1:1 pair-matching ratio without replacement on the logit of the propensity score was employed in this analysis using a caliper of 0.2 width.”

Reviewer #2: 

Thanks for revising the paper extensively. The changes made the paper more insightful and more convincing. The only issue that I'd like to raise is related to Table 1 (page 12).

I think the errors in the table are still uncorrected. The row on nursing school completion above four years says that 14 regular students and 21 quota students took longer than four years to complete school. However, if 150 regular students and 103 quota students graduated, then the percentage taking more than four years should be 9.3% for regular students and 20.4% for quota students.

The errors concern either the percentages OR the numbers in the table note. If the percentages, then also update/correct the statement on page 11 that "there was no significant difference regarding ... the time taken to complete nursing school (p = 0.837)."

Response: Thank you for the comment and careful appraisal of the study. The absolute values of regular and quota students taking more than four years to complete the nursing school were inverted in table 1, and we corrected this issue. The percentage of students taking more than four years to complete the nursing school was 14.0% for regular students (21/150) and 13.6% for the quota students (14/103).

We also changed the statement on page 11, as quoted below. 

“There was no significant difference regarding nursing school attrition (p = 0.629), the time taken to complete nursing school (p = 0.837), and students taking more than four years to complete the nursing school (p = 0.926).”

---

## [Editor Report · Decision Letter 2]

14 Feb 2022

Nursing students admitted through the affirmative action system display similar performance in professional and academic trajectories to those from the regular path in a public school in Brazil

PONE-D-21-11217R2

Dear Dr. Biazotto,

We’re pleased to inform you that your manuscript has been judged scientifically suitable for publication and will be formally accepted for publication once it meets all outstanding technical requirements.

Kind regards,

Giuseppe Lucio Gaeta, Ph.D

Academic Editor

PLOS ONE
---

## [Editor Report · Acceptance letter]

21 Feb 2022

PONE-D-21-11217R2 

Nursing students admitted through the affirmative action system display similar performance in professional and academic trajectories to those from the regular path in a public school in Brazil 

Dear Dr. Biazotto:

I'm pleased to inform you that your manuscript has been deemed suitable for publication in PLOS ONE. Congratulations! Your manuscript is now with our production department. 

Kind regards, 

on behalf of

Prof. Giuseppe Lucio Gaeta 

Academic Editor

PLOS ONE